# Comparative Study of Proximal Femur Bone Tumor Patients Undergoing Hemiarthroplasty versus Total Hip Arthroplasty: A Meta-Analysis

**DOI:** 10.3390/jcm12031209

**Published:** 2023-02-03

**Authors:** Nishant Banskota, Xiang Fang, Dechao Yuan, Senlin Lei, Wenli Zhang, Hong Duan

**Affiliations:** Department of Orthopedics, Orthopedic Research Institute, West China Hospital, Sichuan University, Chengdu 610041, China

**Keywords:** hemiarthroplasty, total hip arthroplasty, dislocation, infection, local recurrence, MSTS, HHS

## Abstract

Hemiarthroplasty and total hip arthroplasty are routinely performed procedures. A comparison of these procedures in tumor patients can be performed based on complications and functional outcomes. To weigh the advantages and disadvantages of both procedures, a comparative study is indeed required to decide which procedure is more beneficial for primary bone tumor patients. The outcomes of proximal femur tumor-resected patients were collected from research reports from PubMed, MEDLINE, EMBASE, Cochrane, and Google Scholar until 30 December 2022. Differences between these two operative procedures in primary bone tumors patients were analyzed based on dislocation, infection, local recurrence, MSTS, and HHS. Six articles were included according to the selection criteria with a total of 360 patients. Our results showed that there was a significant difference in our primary outcome as hemiarthroplasty participants encountered less dislocation than those with total hip arthroplasty. Moreover, the secondary outcomes of our study were similar. Proximal femur bone tumors, when resected, tend to produce more complications and decrease functional ability due to extensive tumor extension and soft tissue involvement. The lower dislocation rate in hemiarthroplasty participants emphasizes the importance of preserving the acetabular head in hemiarthroplasty as a key to preventing dislocation.

## 1. Introduction

Surgical intervention of tumor lesions of the hip is complex and requires major operative management. The proximal femur has been associated with primary bone tumors and metastatic tumors [1]. Before the development of endoprostheses, the primary treatments for these lesions were hip disarticulation or hindquarter amputation [2]. With advances in radiation therapy and chemotherapy, limb salvage became an option in the early 1980s [2]. Pathologic lesions in this important weight-bearing zone weaken its ability to sustain load, causing pain and leading to instability and fracture at this junction [3]. This scenario may have been preluded by biomechanical forces and critical vascularization associated with a poor healing percentage [4,5]. Management of these lesions varies from benign neglect to internal fixation and, recently, to prosthetic reconstruction for optimum function [3]. Reconstructive surgery provides stability and promotes better post-operative rehabilitation. Acetabular component preservation is the most attractive feature of hemiarthroplasty (HA).

Total hip replacement (THA) requires replacement of the femoral head and acetabulum. Compared with THA, HA only replaces the femoral head, which requires less technical skill from the surgeon [6]. The benefits of HA include less surgical trauma, less blood loss, and low economic cost; the disadvantages include a high incidence of postoperative pain and discomfort, and further wear of untreated acetabular cartilage [7,8]. On the contrary, THA surgeries are longer in duration, which ultimately leads to large trauma and heavy blood loss [6]. THA provides pain relief, enhances mobility, and restores function [9]. THA has also been associated with common complications such as aseptic loosening, instability, dislocation, and infection [10]. Both of these procedures have been routinely performed in arthroplasty and trauma departments, with significant positive outcomes demonstrated by various studies [11,12]. Therefore, there is still controversy regarding whether to choose THA or HA for specific patients. Minimal research in this field has been conducted in primary bone tumor surgery. In a study conducted by several authors, contrasting results were found, whereby one study favored THA and the other study favored HA on the basis of the Musculoskeletal Tumor Society (MSTS) scoring system and the Harris Hip Score (HHS) [13,14].

Comparison of these procedures in tumor patients can be measured on the basis of complications and functional outcomes. Dislocation of the hip is a challenging complication following endoprosthesis replacement due to concomitant soft tissue involvement, especially after tumor resection. Endoprosthesis treatment of proximal femur tumor patients in several studies showed conflicting results on the basis of complications and functional outcomes [13,15,16]. In a retrospective study conducted by Varady et al., the authors found no significant difference in short-term morbidity and mortality in HA and THA patients [17]. The outcomes of this study were recorded for up to 30 days and the listed outcomes were major and minor complications, reoperations, readmissions, dependent function status, and local recurrence [17]. To weigh the advantages and disadvantages of both procedures, a comparative study is indeed required to decide which procedure is more beneficial for primary bone tumor patients. We conducted a meta-analysis study in primary bone tumor patients treated with HA and THA on the basis of various outcomes such as complications and functional outcomes.

By searching an abundance of literature on HA and THA treatment in patients with proximal femur tumor lesions, we conducted this meta-analysis to reach a comprehensive conclusion on the post-operative outcomes of these two procedures. These results will help us to guide the optimal management of proximal primary bone tumors located in the proximal hip region. In our study, the HA group was set as the experimental group and the THA group as the control group.

## 2. Methods

This study was performed according to the Preferred Reporting Items for Systematic reviews and Meta-Analyses (PRISMA) guidelines [18]. The PRISMA guidelines chart is presented in the Appendix A.

### 2.1. Literature Search

The MEDLINE, Cochrane, EMBASE, PUBMED, and Google Scholar databases were searched for relevant data until 30 December 2022. The reference lists of relevant studies were also hand-searched by two different authors. Keywords used for searching included primary bone tumors, metastatic tumor, proximal femur tumors, complications, dislocations, local recurrence, infection, functional outcome, HHS (Harris Hip Score), and MSTS (Musculoskeletal Tumor Society). Then, manual searches of the reference lists of the studies found in the databases were conducted.

### 2.2. Included Studies

#### 2.2.1. Inclusion Criteria

English language studies including patients diagnosed with primary bone tumors located in the proximal femur;Studies comparing patients who had undergone THA and HA;Studies that presented comparative data in the respective outcomes of our study.

#### 2.2.2. Exclusion Criteria

4.Non-English and unpublished studies;5.Non-comparative studies in patients undergoing THA and HA with primary bone tumor;6.Case reports, reviews, and letters to editors;7.Studies that lacked adequate clinical data;

Studies that met the criteria for inclusion in the review were not limited to a specific date of publication. Disputes and uncertainties regarding eligibility and viability were resolved through discussions among the reviewers, when necessary.

### 2.3. Study Selection and Data Extraction

Two authors scanned all the abstracts and titles to evaluate whether the studies addressed the questions raised by our study. Outcomes were collected from the articles by two authors of our study. The authors made a structured table, and then, collected all the data, as well as the related information, in a database. The following data were extracted from articles according to the inclusion criteria: the name of the first author, year of publication, study design and protocol, number of patients in each group, patients’ age and gender, dislocation, infection, local recurrence, MSTS, and HHS.

### 2.4. Quality Assessment and Outcome Measurement

Pieces of literature focusing on similar research issues were included, and all studies were retrospective. All studies had low bias as the studies were similar with similar inclusion criteria, surgical procedures, and study periods. Quality assessment was performed using the Newcastle Ottawa Scale (NOS) [19]. In our study, dislocation was set as a primary outcome, and the secondary outcomes were listed as infection and local recurrence, MSTS, and HHS. Dislocation can be defined as an implant or some part of the implant coming out from its originally fixed place, and in general, in hip dislocation, the ball of the implant coming out from the socket.

### 2.5. Statistical Analysis

The outcomes of the measurement used in our study were dislocation, infection, and local recurrence, which were dichotomous data, and MSTS and HHS, which were continuous data. We used the software of the Cochrane Collaboration (Review Manager 5.2) to calculate the odds ratios (ORs) and 95% confidence intervals (CIs) for the dichotomous data and standard mean difference (SMD) and 95% confidence intervals (CIs) for the continuous data. Statistical heterogeneity among the included studies was evaluated using I^2^ tests [20]. Statistically significant heterogeneity was defined as an I^2^ value >0.5 [20]. Heterogeneity was defined as low, moderate, or high based on the I^2^ value (<40%: low; 30–60%: moderate; 50–90%: substantial; >75%: high). I^2^ illustrates the percentage of the total variability in effect estimates among trials that is due to heterogeneity rather than chance [20]. A random-effects model was selected for heterogeneous data; otherwise, a fixed-effect model was selected. Publication bias was detected through funnel plots, which exhibited an intervention effect from the individual study against the respective standard error. An asymmetrical plot suggests that there was no publication bias, and any asymmetry of the plot suggests the existence of publication bias.

## 3. Results

### 3.1. Study Selection

In the primary literature search, 120 relevant articles were retrieved and 62 were excluded based on the exclusion criteria. The abstracts of the remaining 58 were screened, and 30 were excluded based on the exclusion criteria. After conducting all the reviews of the remaining 28 studies, 14 were excluded due to lack of outcome of (*n* = 14) and duplication in the study population with other articles (*n* = 8). Finally, a total of six articles were included in the meta-analysis. The study selection chart is presented in Figure 1. The characteristics of the studies are summarized in Table 1, and the outcomes are summarized in Table 2. 

### 3.2. Dislocation

Only four of the six studies reported dislocation. Dislocation occurred in four of the six studies either in the HA group or the THA group. For this outcome, a fixed-effects model of analysis was used. There was a significant difference in the reported dislocation rate between HA and THA surgery participants undergoing surgery for primary bone tumors, and fewer dislocations were seen in the experimental HA group (OR = 0.42, 95% CI 0.17–1.03, *p* = 0.06), as shown in Figure 2.

### 3.3. Infection

Only two of the six studies reported infection. Infection occurred in two of the six studies either in the HA group or the THA group. For this outcome, a fixed-effects model of analysis was used. There was a significant difference in the reported infection rates between HA and THA surgery participants undergoing surgery for primary bone tumors, and fewer infections were seen in the experimental HA group (OR = 0.73, 95% CI 0.18–2.95, *p* = 0.66), as shown in Figure 3.

### 3.4. Local Recurrence

Only two of the six studies reported local recurrence. Local recurrence occurred in two of the six studies either in the HA group or the THA group. For this outcome, a fixed-effects model of analysis was used. There was a significant difference in the reported local recurrence between HA and THA surgery participants undergoing surgery for primary bone tumors, and fewer local recurrences were seen in the experimental HA group (OR = 0.44, 95% CI 0.09–2.08, *p* = 0.30), as shown in Figure 4.

### 3.5. MSTS

Two studies reported MSTS in our study. For this outcome, a fixed-effects model of analysis was used. There was only a minimal difference in the SMD of MSTS in the operated limbs of HA participants when compared with THA participants undergoing resection for primary bone tumors (SMD = −0.61, 95% CI [−1.04, −0.18], *p* = 0.006), as shown in Figure 5. The MSTS result did not provide any significant values to support either the HA or the THA group.

### 3.6. HHS

Two studies reported HHS in our study. A random-effects model of analysis was used. There was only a minimal difference in the SMD of HHS in the operated limbs of HA participants when compared with THA participants undergoing resection for primary bone tumors (SMD = −0.34, 95% CI [−2.96, 2.28], *p* = 0.003), as shown in Figure 5. HHS result did not provide any significant values to support either the HA or the THA group.

### 3.7. Sensitivity Analysis

Sensitivity analyses indicated that the included studies were performed to determine the reliability of the results, with each study removed in turn. The direction and magnitude of the combined estimates did not change markedly with the exclusion of individual studies, indicating that the findings of the meta-analysis are reliable and suggesting that the results of this meta-analysis are significant and non-fluctuating. Sensitivity analysis was performed for the primary outcome dislocation. The statistical value when the first study was excluded were OR = 0.44, 95% CI 0.17–1.14, and *p* = 0.09); when only the second study was excluded were OR = 0.26, 95% CI 0.09–0.75, and *p* = 0.01; when only the third study was excluded were OR = 0.53, 95% CI 0.16–1.77, and *p* = 0.30; and when the fourth study was excluded were OR = 0.51, 95% CI 0.19–1.37, and *p* = 0.18. All the sensitivity analysis figures are shown in the Appendix A.

### 3.8. Publication Bias

The funnel plot of the dislocation, infection, local recurrence, MSTS, and swing phase are shown in the figure added in the Appendix A. The funnel plot is used for all the outcomes of our study. The findings of the funnel plot show that there is no evidence of publication bias for any of the outcomes.

Overall, in our findings, there was not much heterogeneity in our outcomes; only one outcome, HHS, reported an I^2^ of 89%, in which a random-effects model was selected.

## 4. Discussion

Patients who are candidates for wide and extensive femoral resection because of malignant tumors have long been considered a severe risk group for limb-sparing procedures because of the extent of soft tissue resection and the use of radiation therapy and adjuvant chemotherapy [24]. The decision on the type of surgery must be according to the patient’s estimated residual survival so that whatever the implant and prosthesis type used (bone resection and reconstruction with total joint replacement or intercalary spacer or internal fixation), it will outlive the patient’s life-span [25,26]. Good local control of the tumor and the possibility of better prognosis have enabled many surgeons to perform limb salvage surgery, which also include THA and HA. The intracapsular site of the femur neck makes it biologically possible for tumors of the proximal femur to spread into adjacent anatomical structures such as the hip, synovium, joint capsule, and ligamentum teres [24]. The ligamentum teres provides a mechanism for transarticular skip metastases to the acetabulum. Fortunately, intra-articular involvement is rarely seen and usually occurs after a pathologic fracture [24], and hence, the capsule can be preserved and an intracapsular resection can be conducted [24].

Dislocation is the most common complication encountered after proximal femur resection; rates range from 11% to 15% [27,28]. In our study, we observed fewer dislocations in participants who underwent HA when compared with those who underwent THA (OR = 0.42, 95% CI 0.17–1.03, *p* = 0.06). Dislocation was the primary outcome in a retrospective study conducted by Stevenson et al., and this study was conducted for participants undergoing acetabular replacement for proximal femur tumor resection [29]. The outcome of the 44 participants undergoing HA yielded no dislocation [29]. The studies that are mainly conducted with tumor participants do not have large sample sizes due to the rarity of the tumor. For better justification of our findings, a retrospective study was conducted between 281,140 participants with HA and 5129 with THA; the author found less dislocation in HA patients [30]. The study was conducted by Ogawa et al., on femoral neck fracture participants [30]. A few studies conducted direct comparisons between HA and THA in tumor patients. Contrasting results were observed in one study favoring HA [2] and another study favoring THA [22]; both studies were retrospective. Preservation of the acetabulum and the acetabular articular surface might be an advantage that benefits HA with less dislocation. Another reason for less dislocation in HA is the large femoral head size [3]. The higher rate of dislocation using THA could be the result of implant design as dual-mobility designs and femoral head size were associated with wear and intraprosthetic dislocations [22]. Singh and Bhalodiya showed a significant reduction in the dislocation rate when the size of the head increased in THA participants [31], so newer ideas have been evolving; however, our study still favored HA in preventing dislocation.

All our included studies had a longer follow-up evaluation to justify the reliability of the reported local recurrence rate in our respective studies. Hip arthroplasty prosthetic infection rates of between 1% and 6% have been broadly reported [32,33]. Joint infection is a debilitating complication in itself that may not only require further revision surgery, prolonged hospitalization, and antibiotic treatment, but may also cause tumor patients significant further risks such as amputation or compromised overall survival due to interference with radio- or chemotherapy [34,35]. In our study, authors found less infection in participants undergoing HA, but the findings were not so significant as to distinguish these procedures on the basis of infection. Local recurrence after surgical resection plays a major role in diminished survival and quality of life for primary bone tumor patients [36]. In the context of local recurrence, HA participants reported lower recurrence (OR = 0.44, 95% CI 0.09–2.08, *p* = 0.30), similar to the study conducted by Zucchini et al. [21]. Functional outcomes were evaluated using the MSTS scoring system [37], as was the HHS [38]; in both, a higher score was indicative of good functional mobility. Harris (1969) developed HHS with a rating scale of 100 points and with the domains of pain, function, activity, deformity, and motion [39]. HHS was classified into four subsets, including poor function (HHS < 70), fair function (HHS = 70 to 79), good function (HHS = 80 to 89), and excellent function (HHS = 90 to 100) [40]. The MSTS scoring system developed in 1985 was designed to measure functional outcomes and quality of life after treatment for musculoskeletal tumors, with function also being one of the main components [41]. The six components of MSTS are pain, function, emotional acceptance, lifting ability, manual dexterity, and hand positioning [42]. A comparative study of HHS and MSTS in primary bone tumor patients was conducted by Jamshidi et al. in participants undergoing HA and THA, and this study produces similar results to our study [22]. The MSTS score and HHS are not adequate indicators of quality of life, but in the bone tumor scenario, there has not been much research conducted that focuses on quality of life as the outcome. Our meta-analysis included studies with MSTS and HHS outcomes, all of which had satisfactory MSTS and HHS, with a mean value of almost 70; this might suggest that after surgery, participants had decent quality of life.

A few limitations of this meta-analysis should be illustrated. Firstly, the lack of detailed and verified data from original studies made it hard to adjust estimates by age, menopausal status, lifestyle, smoking, race, etc.; more accurate analysis requires complete data and a large sample size. Secondly, we could not study other important parameters such as aseptic loosening, revision surgery, and functional parameters (TESS). Thirdly, we could not add other studies than retrospective studies such as RCT and prospective ones, as RCT and prospective studies are high-quality and could shed more light on this rare topic; moreover, there is a lower chance of creating bias for the readers. Fourthly, since there were only a few studies published, it was difficult to obtain statistically significant results.

The novelty of this study is in the rarity of this topic as limited research has been conducted in this field on these two routinely performed procedures, for which the advantages and disadvantages against primary bone tumors are studied in this meta-analysis. Other advantages of this meta-analysis are mentioned below. Firstly, a systematic review of the association between two surgical procedures (HA and THA) in primary bone tumor patients is statistically more powerful than any single study. All the studies provided significant comparative data which signified the advantages of prosthesis insertion in proximal femur resection for decreasing complications and improving functions. Secondly, all of the retrospective studies were of high quality and conformed to our inclusion criteria. Thirdly, even though the included studies were few, they still produced statistically significant results, and our study highlights the importance of arthroplasty surgery used in primary bone tumor patients on the basis of complications and limb functions. Therefore, a conclusion has been established that highlights the need for further studies elaborating on its impact on rehabilitation.

## Figures and Tables

**Figure 1 jcm-12-01209-f001:**
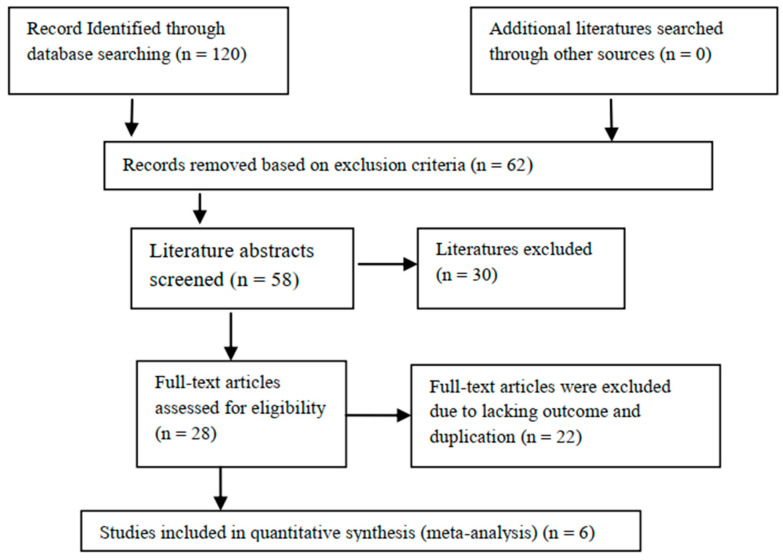
Flow chart of studies included and excluded.

**Figure 2 jcm-12-01209-f002:**
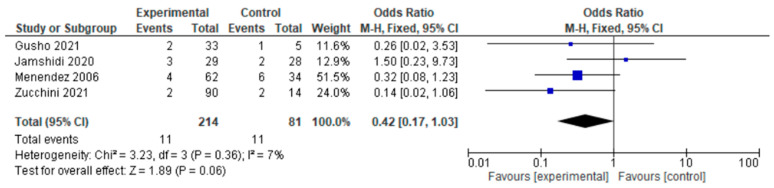
Forest plot of dislocation comparing in proximal femur tumor patients undergoing HA and THA [2,21,22,23].

**Figure 3 jcm-12-01209-f003:**
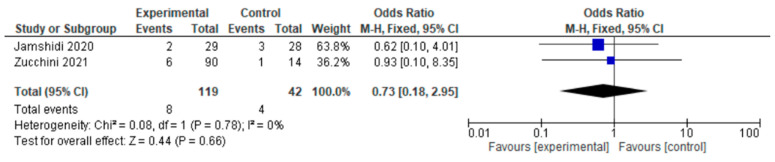
Forest plot of infection comparing in proximal femur tumor patients undergoing HA and THA [21,22].

**Figure 4 jcm-12-01209-f004:**
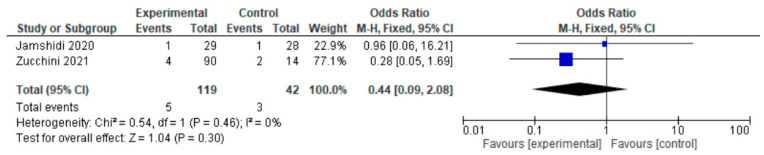
Forest plot of local recurrence comparing in proximal femur tumor patients undergoing HA and THA [21,22].

**Figure 5 jcm-12-01209-f005:**
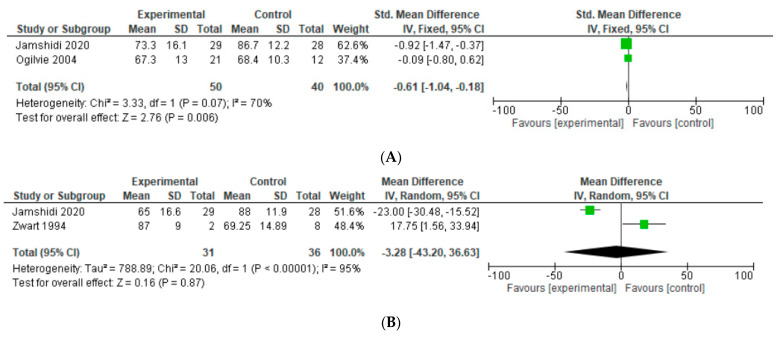
(**A**) Forest plot of MSTS comparing in proximal femur tumor patients undergoing HA and THA. (**B**) Forest plot of HHS comparing in proximal femur tumor patients undergoing HA and THA [13,14,22].

**Table 1 jcm-12-01209-t001:** Characteristics of the included studies.

Studies	Study Period	Patient Number	HA	THA	Male/Female	Median Age	Study Design	Newcastle–Ottawa Quality Score	Country
Zucchini et al., 2021 [21]	NA	104	90	14	51/53	NA	Retrospective	9	Italy
Menendez et al., 2006 [2]	1992–2003	96	62	34	45/51	59	Retrospective	8	USA
Jamshidi et al., 2020 [22]	1998–2016	57	29	28	26/31	33	Retrospective	9	Iran
Ogilvie et al., 2004 [13]	1992–2002	33	21	12	21/12	46.4	Retrospective	8	Canada
Gusho et al., 2021 [23]	2005–2019	38	33	5	15/23	NA	Retrospective	8	USA
Zwart et al., 1994 [14]	1984–1991	33	2	8	15/18	38	Retrospective	8	The Netherlands

NA (not available)—information not available in the respective trial.

**Table 2 jcm-12-01209-t002:** Outcomes of the included studies.

Reference	Dislocation HA, THA	Infection HA, THA	Local Recurrence HA, THA	HHS HA/THA	MSTS HA/THA
Zucchini et al., 2021 [21]	2/90, 2/14	6/90, 1/14	4/90, 2/14	NA	NA
Menendez et al., 2006 [2]	4/62, 6/34	NA	NA	NA	NA
Jamshidi et al., 2020 [22]	3/29, 2/28	2/29, 3/28	1/29, 1/28	65/88	73/86
Ogilvie et al., 2004 [13]	NA	NA	NA	NA	67/68
Gusho et al., 2021 [23]	2/33, 1/5	NA	NA	NA	NA
Zwart et al., 1994 [14]	NA	NA	NA	69/87	NA

NA (not available)— information not available in the respective trial; note: dichotomous data (dislocation, infection, and local recurrence) are reported as the total number of patients with specific outcomes out of the total patients, and the reported continuous data (MSTS and HHS) values are mean values. *p*-values were not been reported in the respective studies, and hence, are not included in this table.

## Data Availability

All the data generated or analyzed during this study are included in the published article.

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
