# Peer review of "Comparative Study of Proximal Femur Bone Tumor Patients Undergoing Hemiarthroplasty versus Total Hip Arthroplasty: A Meta-Analysis"

_jcm, 2023, doi:10.3390/jcm12031209_

Round 1

Reviewer 1 Report

The authors analyzed proximal femur bone tumor patients undergoing Hemiarthroplasty Versus Total hip arthroplasty. The meta-analysis is so interesting, however, I have some concerns to discussed.

-What is the novelty of this study?

-What is the postoperative quality of life?

-Is there any comparison of operative time?

Author Response

Response to Reviewer 1 Comments

Point 1: What is the novelty of this study?

Response 1: The novelty of this study might be the rarity of this topic as limited research conducted in this field as these two procedures are routinely performed, which advantages and disadvantages of primary bone tumors are studied in this meta-analysis. This has been mentioned in lines 365-368.

Point 2: What is the postoperative quality of life?

Response 2: MSTS score and HHS is not adequate indicator of the quality of life, but in bone tumor scenario there have not been much research conducted focusing on the quality of life as its outcome. Our meta-analysis included studies with MSTS and HHS outcomes, all had satisfactory MSTS and HHS with the mean value of almost 70, this might just suggest participants after surgery had a decent quality of life. This part is mentioned in lines 347-352. We could not find studies that directly compared the quality of life as its outcome.

Point 3: Is there any comparison of operative time?

Response 3: We could not find studies that directly compared the operative time of these two procedures. In general concept hemiarthroplasty surgical duration is lower than THA.

Reviewer 2 Report

The manuscript addresses an interesting topic, on which orthopaedic oncological surgeons have not yet reached agreement (THA vs HA in proximal femur replacement).

Given the interest in the topic, the manuscript might be worthy of publication after a revision.

- A complete and accurate check of the bibliography is necessary, because the reference numbers in the text do not seem to correspond with those at the end of the manuscript.

- In table 1 add for each study the total number of patients with THA and patients with HA.

- In the table 2 the data as reported are not easily understood, report the rate of the various complications in terms of percentage in the THA group and the HA group (highlighting the P values).

- Line 67-69 emphasize how this zone has special characteristics both for being a loading zone and for type of vascularization (if appropriate, https://doi.org/10.1186/s12891-022-05728-5).

- A linguistic revision is also necessary to improve the readability of the manuscript.

Author Response

Response to Reviewer 2 Comments

Point 1: A complete and accurate check of the bibliography is necessary, because the reference numbers in the text do not seem to correspond with those at the end of the manuscript.

Response 1: A complete and accurate check of the bibliography has been conducted. The reference at the end of the manuscript corresponds with our manuscript.

Point 2: In table 1 add for each study the total number of patients with THA and patients with HA.

Response 2: Added and highlighted in table 1.

Point 3: In the table 2 the data as reported are not easily understood, report the rate of the various complications in terms of percentage in the THA group and the HA group (highlighting the P values).

Response 3: Some grammatical corrections have been done for better understanding. P – values have not been reported in the respective study and hence not included in the table.

Point 4: Line 67-69 emphasize how this zone has special characteristics both for being a loading zone and for type of vascularization (if appropriate, https://doi.org/10.1186/s12891-022-05728-5).

Response 4: Some points have been added to emphasize this section. Pathologic lesions in this important weight-bearing zone weaken its ability to sustain load causing pain and leading to instability and fracture at this junction. This scenario may have been preluded by biomechanical forces and critical vascularization associated with poor healing percentage

Point 5:  A linguistic revision is also necessary to improve the readability of the manuscript.

Response 5: A major linguistic revision is done and listed below. Revised lines are 39, 55, 63, 76-77, 94-96, 130-131, 153-154, 160, 167-169, 171, 292-295, 306-307, 317, 356-358, 362-363, 371, and 380.

Round 2

Reviewer 1 Report

The authors replied well.

Author Response

Reviewer responded with a “the author replied well” comment, so thanks for the reviewer positive response.

No Comment added to this file.

Reviewer 2 Report

The authors' effort is appreciable in terms of linguistic revision of the manuscript, which certainly appears improved.

But, the other points have not been improved and corrected. 

1- References still do not match between manuscript and bibliography )see number 13 which does not refer to PRISMA guidelines).

2- The data in Table 2 should be reported as the number of patients with the specific complication out of the total (and the P values, calculated and reported in the text in the results section, should be reported in the table).

Author Response

Thanks for the reviewer positive response

Point 1: References still do not match between manuscript and bibliography)see number 13 which does not refer to PRISMA guidelines).

Response 1: This issue has been corrected and a new reference added. Respond to reference 18 line 127.

Point 2: The data in Table 2 should be reported as the number of patients with the specific complication out of the total (and the P values, calculated and reported in the text in the results section, should be reported in the table).

Response 2: Table 2 has been updated with the outcome reported as number of the patients with the specific complication out of the total. But we could not add the P-value as we could not find P-value in the related article and even though we had sent an email request for the availability of missing data to the corresponding author we did not get any positive response. We are trying to improve this manuscript for a better understanding of the readers.
